engineering geology

tailings, basalt fibre reinforcement, strength, compression index, permeability coefficient, sensitivity

**Author for correspondence:**
Dongming Zhang
e-mail: zhangdm@cqu.edu.cn

# Mechanical and permeation response characteristics of basalt fibre reinforced tailings to different reinforcement technologies: an experimental study

Jianzhong Liu[1,2,3], Han Yang[1,2], Dongming Zhang[1,2], Yun Wang[4], Weijing Xiao[1,2], Chen Ye[1,2], Binbin Zheng[5] and Yushun Yang[6]

[1]State Key Laboratory of Coal Mine Disaster Dynamics and Control, and [2]School of Resources and Safety Engineering, Chongqing University, Chongqing 400044, People's Republic of China
[3]China Coal Technology and Engineering Group, Beijing 100013, People's Republic of China
[4]China Railway 23rd Bureau Group 6th Engineering Co. Ltd, Chongqing 401121, People's Republic of China
[5]School of Management Science and Engineering, Shandong Technology and Business University, Yantai 264005, Shandong, People's Republic of China
[6]Faculty of Architecture and Civil Engineering, Huaiyin Institute of Technology, Huai'an 223001, Jiangsu, People's Republic of China

 DZ, 0000-0003-0409-3657; YW, 0000-0002-8372-9571; WX, 0000-0002-1524-3414; CY, 0000-0003-3528-5870; YY, 0000-0002-8510-7194

Tailings dam is a man-made hazard with high potential energy; dam failure would cause great losses to human lives and properties. However, the limitations of conventional reinforcement methods like geosynthetic make it easy to slide along the weak structural plane. In this paper, we innovatively added basalt fibre (BF) with different lengths ($l$) and contents ($\omega$) into tailings to study its mechanical and permeation characteristics. The results indicate that BF can improve the shear strength ($\tau$), cohesion ($c$) and compression index ($C_c$) of tailings, but it has little effect on internal friction angle ($\varphi$). When $l$ is constant, $\tau$, $c$ and $C_c$ are positively correlated with $\omega$. One notable phenomenon is that $\tau$ and $c$ do not constantly increase with $l$ when $\omega$ is constant, but obtain the maximum under the optimal length of 6 mm. Moreover, when $\omega > 0.6\%$, permeability coefficient ($k$) is greater than that of the original tailings and the sensitivity of

$c$, $\varphi$, $\tau$, $C_c$, $k$ to fibre content is greater than that of length. The research results facilitate the understanding of BF reinforced tailings, and could serve as references for improving the safety of tailings dam and other artificial soil slopes or soil structures.

# 1. Introduction

With the rapid development of the global economy, the consumption of resources becomes ever larger. Thus the demand for resources is also growing fast, and mining activities are correspondingly growing larger and larger in scale worldwide. Mineral processing is a vital step in mining engineering; tailings are the waste materials left after the target mineral is extracted from ore, which consist of crushed rock, water, trace quantities of metals and additives used in processing. Tailings, most often in the form of a wet slurry, are conventionally stored above-ground behind tailings dams. It is estimated there are around 18 000 tailings dams globally. China alone has approximately 7800 tailings dams among them, ranking first in the world in total [1–3]. More and more low-grade ores would be mined and much more tailings would be created in the near future given the limited mineral resources. Tailing is usually stored in tailings pond and tailings dam is used to prevent tailings loss. Due to the gradual accumulation of tailings, the tailings dam will become higher and higher, finally forming a dangerous source with high potential energy. Brumadinho tailings dam (in Brazil) collapsed in 2019 and caused 259 deaths and 11 missing [4]; in 2015, Fundao dam broke down and released 34 million m$^3$ mud in short time, triggering the destruction of agricultural areas even native flora of the Atlantic Forest [5]; the Imperial Metals Mount Polley gold and copper mine tailings dam disaster in 2014 dumped 24 million m$^3$ mine waste and sludge into a lake, causing serious environmental pollution [6]. These painful lessons illustrate that tailings dam break may cause heavy casualties and property losses; therefore, it is particularly crucial to improve the stability of the tailings dam and keep its safe operation.

Fluet *et al.* [7] introduced geosynthetics into geotechnical engineering to improve mechanical properties. Other scholars try to apply geosynthetics to tailing pond engineering and obtain many achievements. The main method is to lay geotextile, geogrid, geomembrane and other materials in tailings dam to improve tailings mechanical properties, so as to improve the safety of the tailings pond. Yi & Du [8] found the strength of reinforced tailings increased substantially through triaxial compression tests on geogrid or geotextile-reinforced tailings; besides, pseudo-cohesion of shear strength index increased, whereas the friction angle remains primarily unchanged with the increase in reinforced layers. Ozhan & Guler [9] found that geomembrane-laminated geosynthetic clay liner had both perfect mechanical and hydraulic performance against sliding along the terraced slopes of a boric acid tailings dam. Palmeira *et al.* [10] presented a study on the use of nonwoven geotextiles in drainage and filtration systems of tailings dams and found the overall performance of the geotextiles tested under laboratory conditions was satisfactory. Zhang *et al.* [11] found sinking displacement and collapse were reduced after adding geotextile grid and the stability of the tailings dam was effectively improved through numerical simulation analysis. Yang *et al.* [12] pointed out that the application of geotextile tubes in the construction of tailings dam was beneficial for fine tailings disposal through a case study.

With increasing geotechnical engineering activities, geosynthetics can no longer meet the growing demands. French scientist Henri Vidal put forward reinforced earth theory in the 1960s [13,14]. The theory holds that thanks to the friction between earth grains (including all particle sizes) against the reinforcing members (which can withstand major tensile stress), the cohesion of earth can be improved through introducing reinforcement components. Thereafter worldwide researchers had expanded the application to many aspects of production to study the reinforcement technology. In terms of reinforcement materials: natural fibre and industrial by-products [15], mining waste [16], polypropylene [17], polyacrylonitrile and glass fibres [18] and basalt fibre (BF) [19] had been studied. In terms of reinforcement objects: cemented tailings backfill [20,21], concrete slabs [22], tailings–cement blends [23], soil [24] and seawater sea sand concrete [25] had been investigated. In terms of research methods: triaxial tests [26], unconfined compressive strength [27,28], direct shear tests [29], Brazilian indirect tensile [30], centrifuge tests [31], X-ray diffraction [32], computed tomography [33] and scanning electron microscopy (SEM) [34] had been analysed.

Although conventional geosynthetics can improve the mechanical properties of tailings to some extent, this kind of geosynthetics form weak structural planes between tailings and reinforcement materials, which makes it easy to slide along the weak structural plane and leads to damage

[35,36]. By analysing the existing literature, although fibre has the advantages of high strength, corrosion resistance and good dispersion, we find that there are few reports on fibre reinforced tailings. Hence we innovatively try to add fibres into tailings to explore whether it is technically feasible to improve the natural properties of tailings. Polypropylene fibre, glass fibre and polyacrylonitrile fibre are commonly used reinforcement materials, but the density of polypropylene fibre is less than that of water, so it is not suitable for tailings reinforcement. Glass fibre is fragile, has poor wear resistance and not easy to disperse. Polyacrylonitrile fibre has low strength, poor wear and fatigue resistance and is not suitable either. Nevertheless, basalt fibre (BF) has excellent tensile strength and elastic modulus, good corrosion resistance and chemical stability [37], so BF is selected as tailings reinforcement material in this work. This paper will comprehensively study the influence of BF on strength, consolidation and permeation characteristics of tailings, and analyse its internal mechanism. The aim is to explore the possibility of using BF reinforcement to improve the natural properties of tailings, so as to seek another proper method to enhance the safety of tailings dam.

# 2. Experimental methods

In this section, the influence of BF on tailings' mechanical and hydraulic behaviours will be investigated experimentally by conducting laboratory tests first. Subsequently, the intrinsic mechanism of basalt fibre reinforced tailing (BFRT) will be further analysed by micro imaging techniques using SEM.

## 2.1. Experimental materials

### 2.1.1. Tailings

The tailings sample comes from a tailings pond in Yunnan Province, China. It contains three types of tailings: polymetallic ore tailings, tin copper sulfide ore tailings and oxidized ore tailings, and the share is 18.80%, 50.81% and 30.39%, respectively. The particle size distribution curve was obtained through a Microtrac S3500 laser particle size analyser (figure 1); the physical properties are shown in table 1.

### 2.1.2. Basalt fibre

In this study, BF with different lengths are selected to add into tailings (figure 2). In order to reduce experimental error, BF should be manually separated into monofilaments before adding into tailings when preparing the experimental samples. To make them evenly disperse in tailings, the BF monofilaments could be mixed with tailings first, and then adding the appropriate amount of water for sample preparation. The physical and mechanical parameters of BF are presented in table 2.

## 2.2. Experimental scheme

Direct shear test, consolidation test, permeation test and SEM experiment were applied in this study. The $\tau$, $c$ and $\varphi$ can be obtained by direct shear test according to Coulomb's law. The experimental device for the direct shear experiment is ZJ series strain-controlled direct shear apparatus (by Nanjing Soil Instrument Factory Co. Ltd, Nanjing, China), and four parallel specimens were made under each experimental condition. The instrument for the consolidation test is WG single-lever consolidometer (by Nanjing Soil Instrument Factory Co. Ltd, Nanjing, China). Two parallel samples were made for each experimental condition and the average value was taken as the final $C_c$ of BFRT. The $k$ of BFRT is measured by changeable water pressure permeation test in this work; five groups of $k$ should be measured for each specimen to reduce experimental error, and finally taking the average within the allowable error range (i.e. no more than $2 \times 10^{-n}$, where $n$ is the number of experiments) as the final $k$. Moreover, the mixture of tailings and fibres was scanned by SEM to observe the details to study the intrinsic mechanism of BFRT. It should be noted that Vaseline or other lubricant ought to be coated inside the ring knife before preparing the specimen to reduce the friction. The specific experimental schemes are shown in table 3 (note: fibre content of zero represents no fibre is added into tailings).

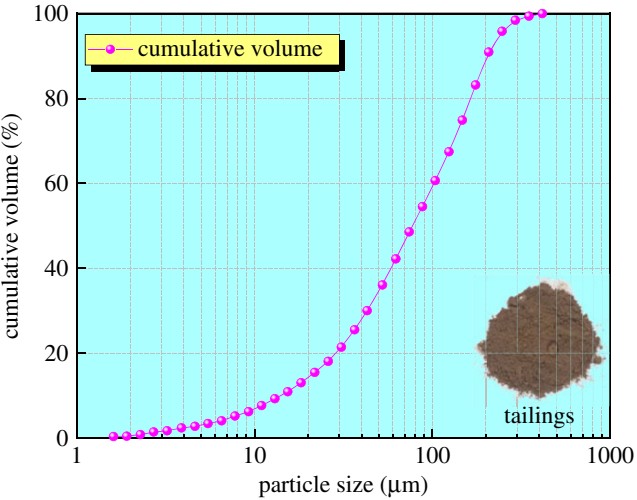

**Figure 1.** Particle size distribution curve of tailings.

**Table 1.** Tailing physical properties.

| specific gravity | plastic limit (%) | liquid limit (%) | plastic index | non-uniform coefficient | curvature coefficient |
|---|---|---|---|---|---|
| 3.03 | 10.01 | 18.81 | 8.80 | 8.94 | 1.40 |

**Table 2.** Physical and mechanical parameters of BF.

| density (g cm$^{-3}$) | tensile strength (MPa) | fracture strength (MPa) | elastic modulus (GPa) | acid and alkali resistance (%) | breaking elongation (%) |
|---|---|---|---|---|---|
| 2.7 | 2650 | 3200 | 89 | 75 | 3.1 |

# 3. Results and discussion

## 3.1. Strength characteristics of BFRT

### 3.1.1. Shear strength analysis

One of the major reasons of tailings dam break is when the shear stress exceeds its shear strength. Consequently, shear strength is a crucial factor affecting the stability of tailings dam. The growth of shear strength helps to boost tailing dam ability to resist failure to some extent [38]. Based on the experimental results, the curve of shear stress $\tau$ to shear displacement $\Delta l$ of BFRT is acquired. The maximum shear stress of $\tau$–$\Delta l$ curve is taken as the shear strength, and when there is no maximum, the corresponding shear stress when shear displacement equals 4 mm is taken as the shear strength (this is the case in this study). According to Coulomb's law that $\tau = \sigma \tan \varphi + c$, the shear strength curve $\tau$–$\sigma$ can be plotted with $\tau$ as the ordinate and the vertical stress $\sigma$ as the abscissa; hence $c$ and $\varphi$ of BFRT can be calculated accordingly. Due to the limitation of article length, only the $\tau$–$\Delta l$ curves of fibre length of 9 mm and fibre content of 0.9% are listed in figure 3.

From figure 3a we conclude that the shear stress of BFRT increases with the increase of fibre content under the same fibre length. When $\Delta l \geq 2$ mm, the variation rate of $\tau$ gradually slows down under various fibre parameters. Figure 3b shows that when fibre content is constant, the shear stress of BFRT does not constantly rise with fibre length but elevates first and then shrinks, and it obtains the maximum at $l = 6$ mm. The detailed reasons will be analysed in the following sections.

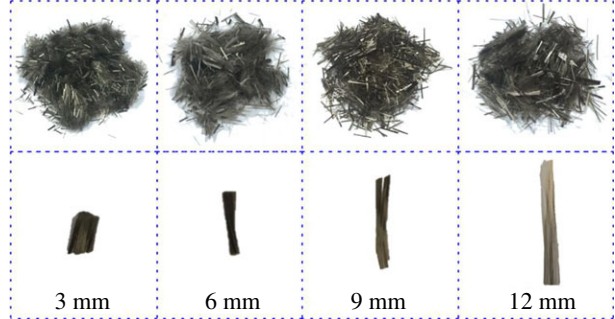

**Figure 2.** Basalt fibre under different lengths.

**Table 3.** Experimental scheme.

| group number | fibre length (mm) | fibre content (%) | group number | fibre length (mm) | fibre content (%) |
|---|---|---|---|---|---|
| 1 | 3 | 0.6 | 10 | 9 | 0.6 |
| 2 | | 0.9 | 11 | | 0.9 |
| 3 | | 1.2 | 12 | | 1.2 |
| 4 | | 1.5 | 13 | | 1.5 |
| 5 | 6 | 0.6 | 14 | 12 | 0.6 |
| 6 | | 0.9 | 15 | | 0.9 |
| 7 | | 1.2 | 16 | | 1.2 |
| 8 | | 1.5 | 17 | | 1.5 |
| 9 | | 0 | | | |

The shear strength of BFRT under different fibre length and content could be obtained according to the curves of shear stress and shear displacement. Experimental results indicate that the addition of BF can improve tailings shear strength. However, when fibre content or length exceed certain thresholds, the shear strength will decrease. Therefore, in this work, the fibre length corresponding to the maximum shear strength of BFRT under each content is defined as 'optimal length' and the content corresponding to the maximum shear strength under each length is defined as 'optimal content'.

Shear strength increment of BFRT was calculated for illustration (figure 4). Figure 4b shows $\tau$ is positively correlated with $\omega$ under the same fibre length, and $\tau$ obtains the summit at $\omega = 1.5\%$ in this work. As shown in figure 4a, $\tau$ increases with the increase of $\omega$ when $l = 3, 12$ mm and acquires the peak at $\omega = 1.5\%$. When $l = 6$ mm, the increment reaches 42.49% when $\omega = 1.5\%$, which suggests the most significant change in this study. When $l = 6$ mm, the increment expresses sharp increase at $\omega = 1.2\%$, which is near twice that when $\omega = 0.6$ and 0.9%. To sum up, $\tau$ manifests even increment at $l = 3, 12$ mm but shows obvious distinctions at $l = 6, 9$ mm.

Moreover, when $\omega = 0.6, 0.9$ and 1.5%, $\tau$ rises to summit first and then decreases with increasing $l$, obtaining the maximum at $l = 6$ mm (figure 4a). However, when $\omega = 1.2\%$, $\tau$ increases monotonically with increasing $l$, and the increment demonstrates the greatest value in the range of 6–9 mm. Consequently, the maximum economic benefit can be achieved with the least fibre cost through selecting 6–9 mm fibres for reinforcement when $\omega = 1.2\%$. In summary, the optimal length is concluded at 6 mm and the optimal content is concluded at 1.5%, the shear strength increasing by 42% under this condition.

### 3.1.2. Cohesion analysis

According to experimental results, $c$ and $\varphi$ could be calculated correspondingly, which are basic indexes to shear strength and have vital impacts on tailings strength.

Figure 5 demonstrates the variation curve of BFRT cohesion with fibre length and content; it shows that $c$ rises significantly after adding BF. The curve of $c$ and $l$ is a parabola, which conforms to quadratic function

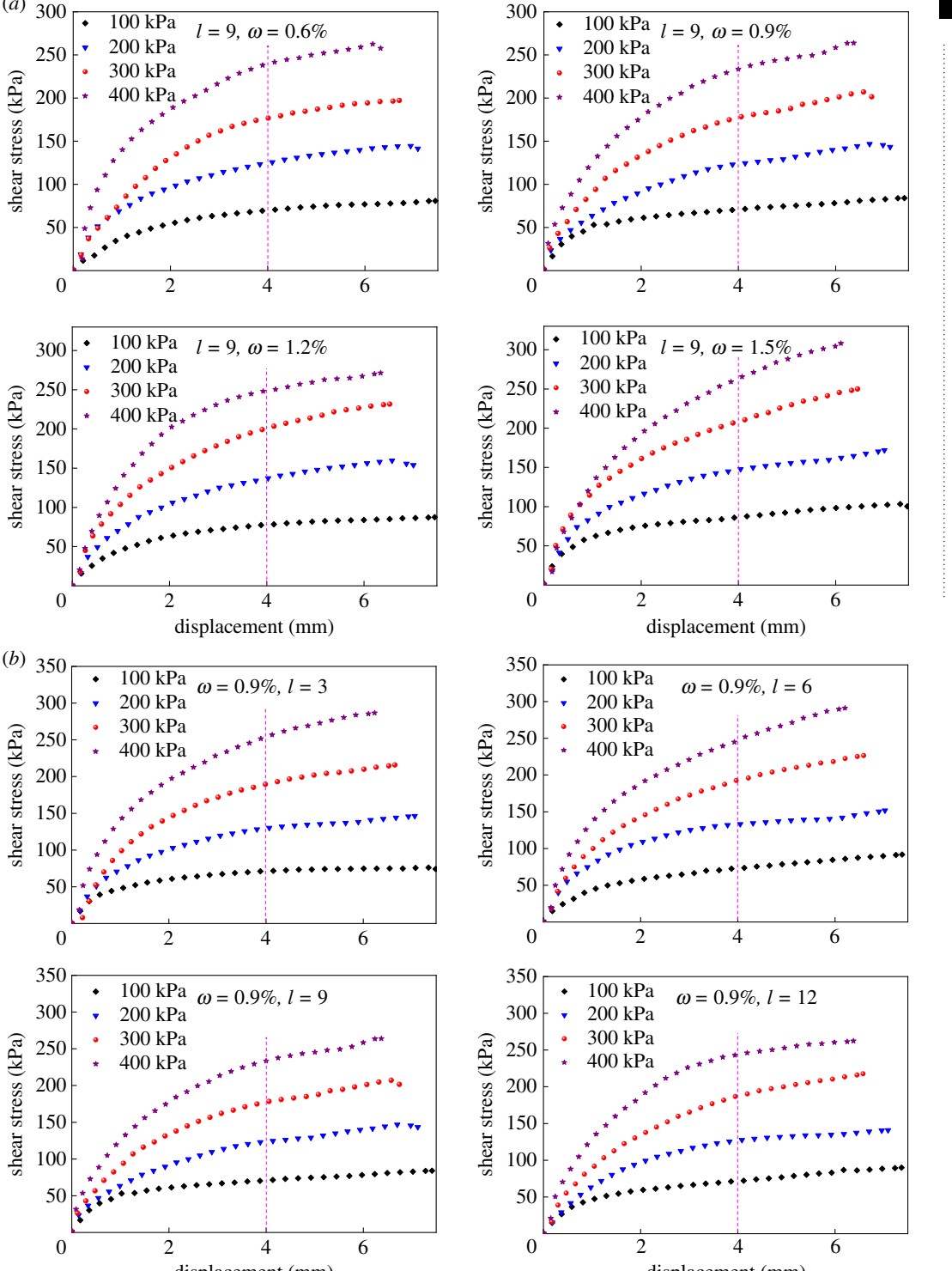

**Figure 3.** $\tau$–$\Delta l$ curves of BFRT: (a) $\tau$–$\Delta l$ curve with fibre length of 9 mm and (b) $\tau$–$\Delta l$ curve with fibre content of 0.9%.

$y = ax^2 + bx + c$, and the curve of $c$ and $\omega$ conforms to exponential function $y = \exp(ax^2 + bx + c)$. Figure 5a implies $c$ increases first and then decreases with increasing $l$. When $\omega = 0.6$, 0.9%, the maximum increment is acquired at $l = 6$ mm, which increase by 136.83% and 182.86%, respectively. However, when $\omega = 1.2$, 1.5%, $c$ acquires the peak at $l = 9$ mm, the increments being 305.74% and 449.79%, respectively. Figure 5b denotes that $c$ ascends continuously with $\omega$ under the same fibre length, reaching minimum and maximum at $\omega = 0.6\%$ and 1.5%, respectively. When $\omega$ varies from 1.2% to 1.5%, the effect is the most significant. By and large, the increment of $c$ reaches the minimum and maximum at $l = 3$ mm and $l = 9$ mm, respectively, but there is no significant difference between $l = 6$ mm and $l = 9$ mm.

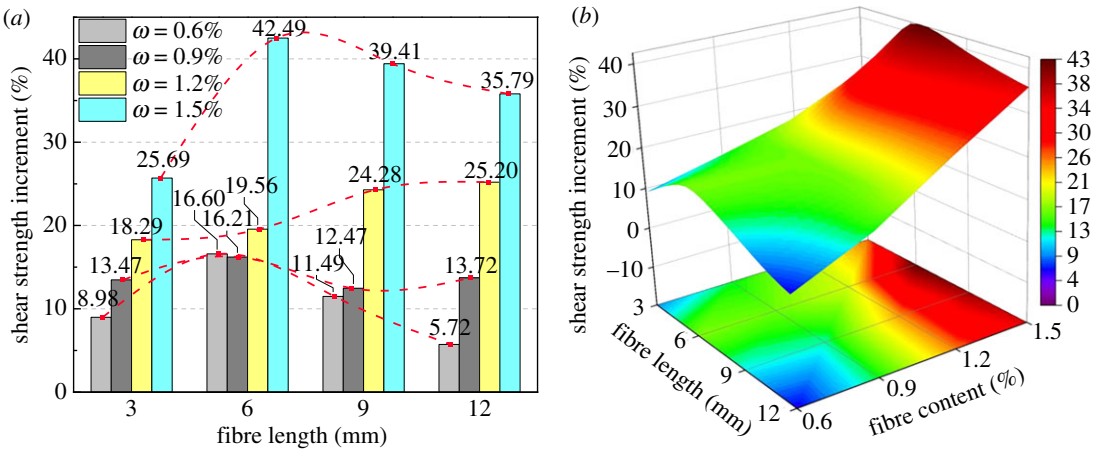

**Figure 4.** Increment of shear strength: (*a*) column and (*b*) contour map.

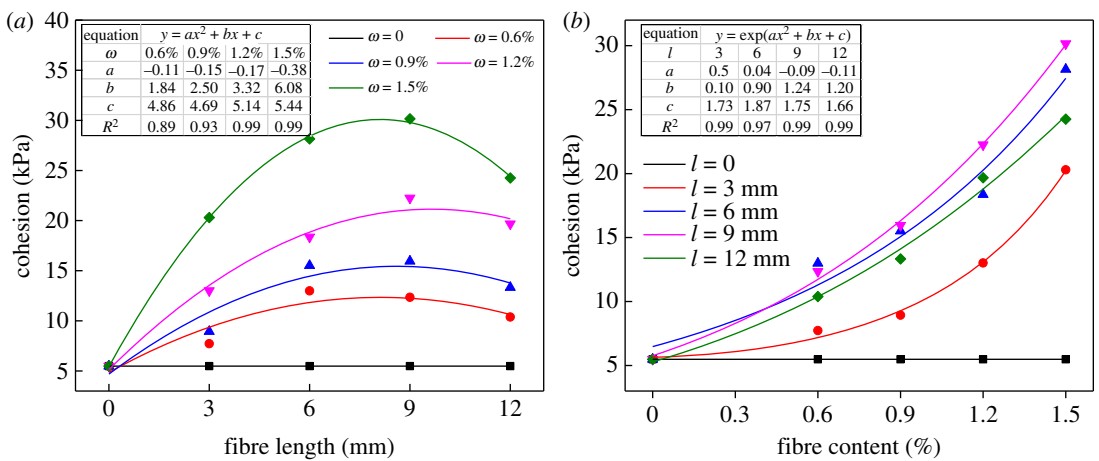

**Figure 5.** Effects of fibre length and content on the cohesion of tailings: (*a*) fibre length and (*b*) fibre content.

Generally, the variation trend of cohesion is similar to shear strength. The cohesion is positively correlated with fibre content, and the optimal fibre length that makes cohesion the greatest is 6–9 mm.

### 3.1.3. Internal friction angle analysis

The light yellow plane represents the $\varphi$ of original tailings (figure 6). The variation range of $\varphi$ is less than 5%, which shows that BF has little influence on $\varphi$. The internal friction angle becomes smaller under some conditions, but in other cases, it becomes larger. $\varphi$ ascends when $l \leq 6$ mm but descends when $l > 6$ mm. The variation law of $\varphi$ is very complex and does not share the same trend. Nevertheless, it is worth noting that when $\omega = 1.5\%$, $\varphi$ is constantly larger than original tailings ignoring fibre length and acquires the maximum at $l = 6$ mm. When $l = 6$ mm, $\varphi$ is invariably larger than original tailings ignoring fibre content.

## 3.2. Consolidation characteristics of BFRT

The increase of compressibility contributes to heighten the effective stress of tailings particle skeleton, and it is profitable to enhance the ability of tailings dam to resist failure. The compressibility of tailings could be expressed by the relation of porosity and vertical stress, and the consolidation curve $e$–lg$P$ is approximately a straight line when vertical pressure is large. Hence the slope of the line is nearly constant and the slope represents the compression index. Compression index can be calculated by the following equation:

$$C_c = -\frac{\Delta e}{\Delta(\lg P)},$$

(3.1)

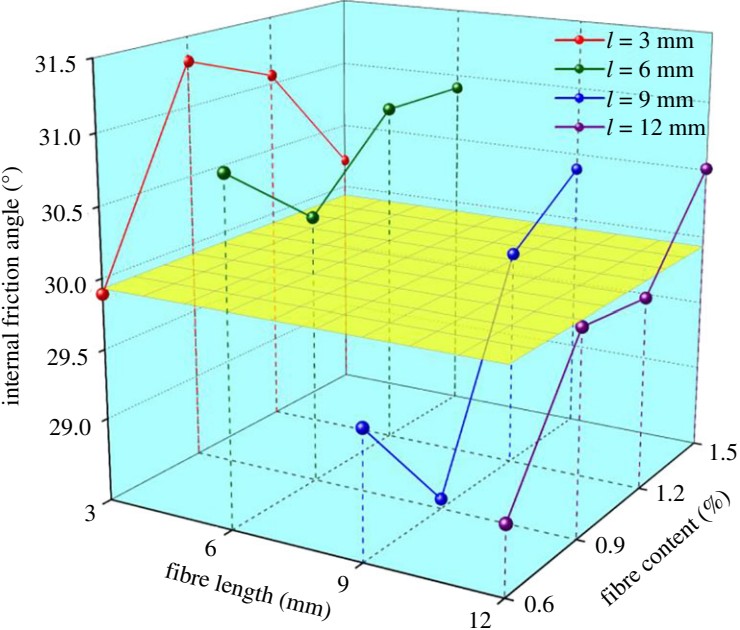

**Figure 6.** Effects of fibre length and content on the internal friction angle of tailings.

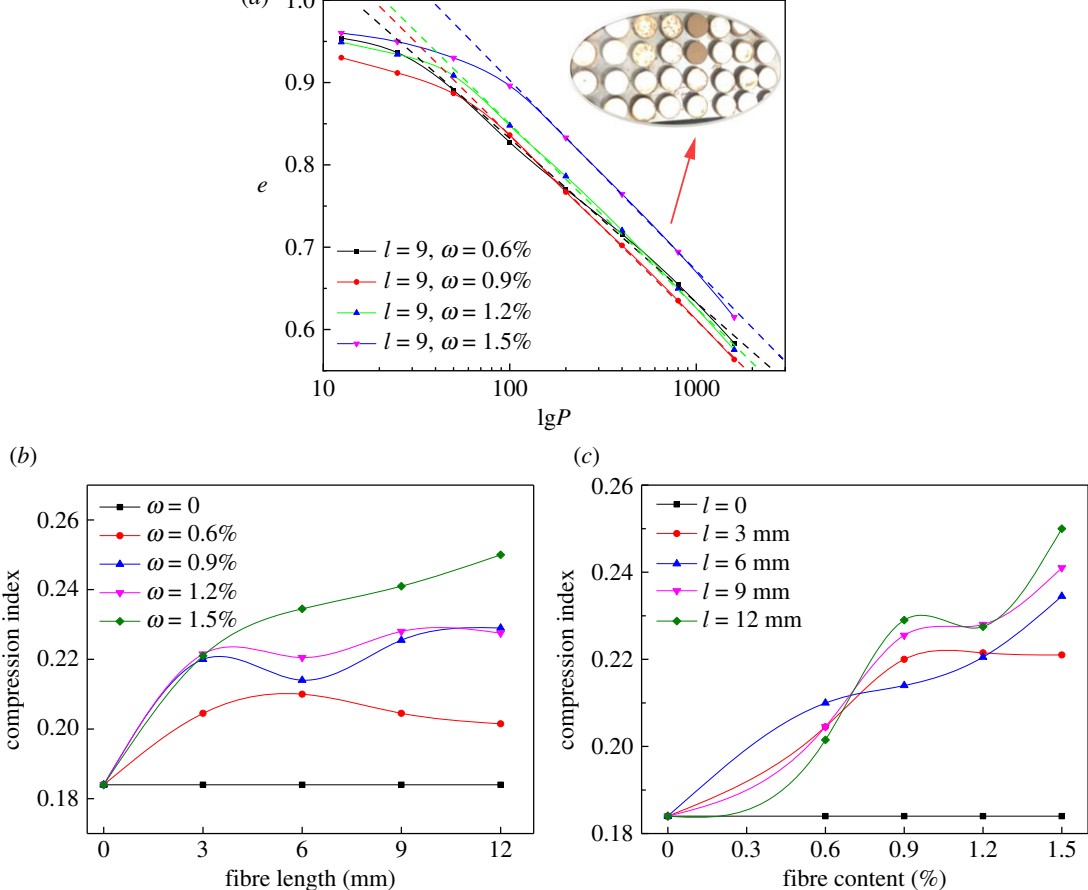

**Figure 7.** Consolidation test curves: (a) $e$–lg$P$, (b) $C_c$–$l$ and (c) $C_c$–$\omega$.

where $e$ is the porosity of specimen; $P$ is the vertical stress (kPa). The $e$–lg$P$ curve of BFRT under fibre length of 9 mm is shown in figure 7a. The compression index under different fibre parameters was then calculated, and the variation curve of compression index with fibre length and content is shown in figure 7b,c.

The porosity declines with increasing vertical stress (figure 7a), which is because the rising pressure makes the tailings particles in the sample get closer. Figure 7b shows the addition of BF has enhanced $C_c$ markedly; when $\omega = 0.6\%$, $C_c$ ascends to the maximum first and then descends, obtaining the maximum at $l = 6$ mm. When $\omega = 0.9$, 1.2 and 1.5%, $C_c$ fluctuates slightly with the increase of $l$ but shows an upward trend in general. As shown in figure 7c, $C_c$ generally expresses an upward trend with $\omega$, but there are still some subtle differences between different fibre lengths. When $l = 3$ mm, the growth rate of $C_c$ gradually slows down with $\omega$ and obtains the maximum at $\omega = 1.2\%$. $C_c$ rises up continuously with increasing $\omega$ when $l = 6$, 9 mm, and fluctuates slightly at $l = 3$, 12 mm. $C_c$ obtains the maximum under $\omega = 1.5\%$ and $l = 12$ mm, when the increment is 35.87%.

Through the above analysis and discussions, we can conclude that adding BF into tailings can improve its compression index, and $C_c$ is overall positively correlated with $\omega$, yet the change is not significant when fibre content is constant. According to table 1, the non-uniform coefficient of tailings is 8.94 and the curvature coefficient is 1.4. Thereby it is well-graded and the fine particles are easy to fill into the pores between the coarse-grained particles during consolidation. However, due to the rough surface and various shapes of tailings particles, the fine particles cannot completely fill the gaps between coarse-grained particles, but the diameter of BF monofilament is smaller than that of tailings particles, hence it is easier to fill the gaps between tailings particles, which contributes to reduce the porosity and promote compression index. Further, when fibre length is constant, greater content means there are more fibres per unit volume, which makes more pores filled with fibres; therefore, $C_c$ rises with increasing $\omega$. The longer the fibre is, the greater the monofilament mass is. Thereby the amount of fibres per unit volume declines with increasing fibre length under the same content. However, the decrease is not significant in the macro view, thus fibre length variation has little influence on $C_c$ when content is constant. The compressibility of tailings increases after adding BF, which leads to the decline of porosity and elevation of effective stress of tailings particle skeleton. It can effectively improve tailings strength, which is beneficial to enhance the strength of tailings dam in practice and improve its safety.

## 3.3. Permeation characteristics of BFRT

Normal and stable seepage could accelerate the formation of dry beach and the consolidation of tailings, correspondingly improving the stability and safety of the tailings dam. If it is not designed and constructed reasonably, the saturation line of the tailings dam would be high, which may lead to dam failure accident. Thereby heightening the permeability coefficient in reasonable range helps to ensure the stability of the tailings dam. The calculation method of permeability coefficient in this work is as follows:

$$k_T = \frac{2.3aL\lg(h_1/h_2)}{A(t_2 - t_1)}. \tag{3.2}$$

The permeability coefficient should be converted to that of standard temperature (20°C) for comparative analysis, and it can be expressed as

$$k_{20} = k_T \frac{\eta_T}{\eta_{20}}, \tag{3.3}$$

where $k_T$ is the permeability coefficient at $T$°C, cm s$^{-1}$; $a$ is the sectional area of changeable water head pipe, cm$^2$; $L$ is the height of specimen, cm; $t_1$ and $t_2$ are the start and end time of counting, respectively, s; $h_1$ and $h_2$ are the start and end water head, respectively, cm; $k_{20}$ is the permeability coefficient at 20°C cm s$^{-1}$; $\eta_T$ and $\eta_{20}$ are dynamic viscosity coefficient of water at $T$°C and 20°C, respectively, kPa s.

The permeability coefficient of tailings is affected by BF (figure 8). When $\omega = 0.6\%$, $k$ is consistently less than that of original tailings ignoring the length (figure 8a). It acquires the minimum at $l = 3$ mm and the shrinkage gradually decreases when $l > 6$ mm. $k$ is always greater than that of original tailings when $\omega = 0.9$, 1.2 and 1.5%. The curve of $k$ is overall in the shape of a parabola, and reaches the peak at $l = 6$ mm. Figure 8b denotes the curve of $k$ under $l = 3$ mm is similar to that of $l = 12$ mm. $k$ descends at $\omega = 0.6\%$ and reaches the summit at $\omega = 0.9\%$. When $l = 6$ mm, $k$ decreases to the minimum first (at $\omega = 0.6\%$) and elevates monotonically with $\omega$ when $\omega > 0.6\%$. The greatest increment is 42.57% at $\omega = 1.5\%$, which is remarkable.

The experiment results suggest that 0.6% is the 'critical content' which leads to the variation of tailings permeability coefficient. The $k$ is consistently less than that of original tailings when $\omega = 0.6\%$,

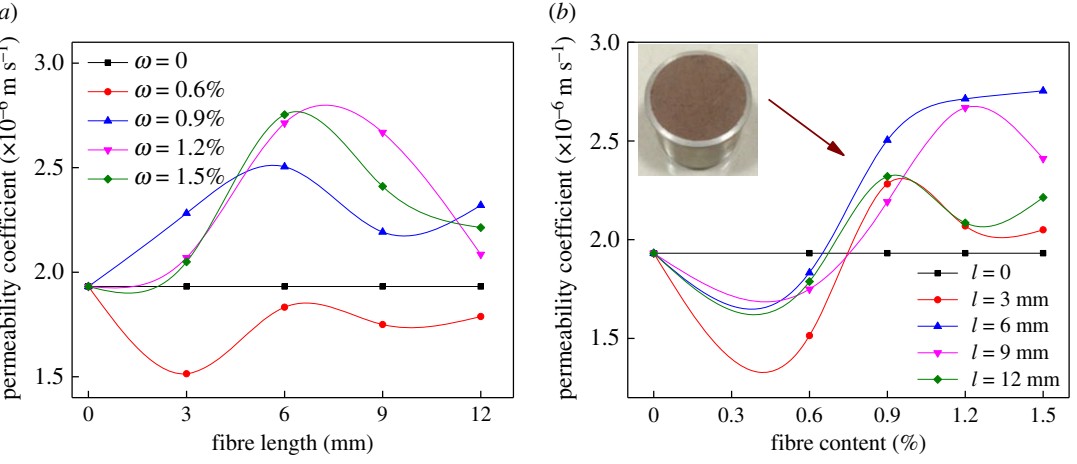

**Figure 8.** Effects of fibre length and content on the permeability coefficient of tailings: (a) fibre length and (b) fibre content.

and it is always greater than original tailings when $\omega > 0.6\%$. The reason why $k$ becomes smaller after adding fibres of dosage of 0.6% may be that original tailings have permeability; when fibre content is small, the fibres fill the pores between tailings particles and block the permeation channel, thus leading to reduction of $k$. When fibre content gets larger ($\omega > 0.6\%$), the amount of fibre between tailings particles gets greater, thus the probability that fibre distribution is consistent with the permeation direction becomes larger. Additionally, the size of BF monofilament is much larger than those tailings particles, and the fibres run through tailings particles in the sample to form a seepage channel, resulting in the rise of permeability. When $\omega > 0.6\%$, $k$ obtains the summit value at $l = 6$ mm. The reason may be that BF is too short to produce a marked channelling effect when $l < 6$ mm. However, when $l > 6$ mm, BFs form into a net shape and some of the monofilaments bend in the specimen due to the limited size of the sample, resulting in the length of the seepage channel being actually less than fibre length, so the fibre with a length of 6 mm is the most appropriate. In practice, the elevation of permeability helps to reduce the saturation line height of the tailings dam, which plays a critical role in the safe operation of tailings dam.

## 3.4. Sensitivity analysis

In order to figure out the responsiveness of $\tau$, $c$, $\varphi$, $C_c$ and $k$ to the change of fibre length and content, sensitivity analysis is carried out in this section. Sensitivity is expressed as

$$S_i = \left| \frac{(\Delta Y_i / Y_i)}{(\Delta x_i / x_i)} \right|, \tag{3.4}$$

where $S_i$ is the sensitivity, $\Delta Y_i / Y_i$ is the relative change rate of $\tau$, $c$, $\varphi$, $C_c$ and $k$, and $\Delta x_i / x_i$ is the relative change rate of influencing factors. The larger $S_i$ is, the more sensitive the index is. Since the main variables in this experiment are fibre length and content, thereby fibre length and content are determined as the main influencing factors.

The red line represents the sensitivity of $\tau$, $c$, $\varphi$, $C_c$ and $k$ to fibre content under different lengths, and the brown one signifies the sensitivity of $\tau$, $c$, $\varphi$, $C_c$ and $k$ to fibre length under different contents (figure 9). Figure 9a,b shows that the sensitivity of $\tau$ and $c$ increases as BF content increases under various fibre lengths, and it decreases with increasing length under different fibre contents. From figure 9c,d we can conclude that the sensitivity of $\varphi$ and $C_c$ is small to BF length, namely, BF length change has little effect on $\varphi$ and $C_c$. Figure 9e indicates that the sensitivity of $k$ to BF length and content decreases with increasing length and content. In short, $\varphi$ expresses the lowest sensitivity to BF length and content change, while $c$ expresses the highest sensitivity. The sensitivity is ranked as follows: $c > k > \tau > C_c > \varphi$; that is to say, adding BF has the greatest effect on $c$ and the least effect on $\varphi$, which is consistent with the previous analysis. Furthermore, figure 9 also shows that the sensitivity of $\tau$, $c$, $\varphi$, $C_c$ and $k$ to content is greater than that to length. Therefore, it is more effective to improve tailings mechanical properties by altering BF content than length in practice.

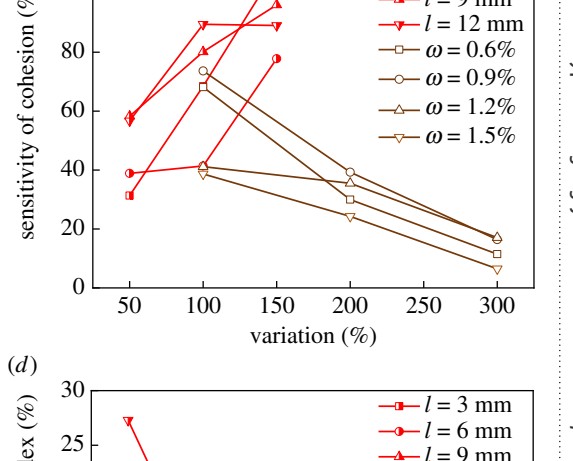

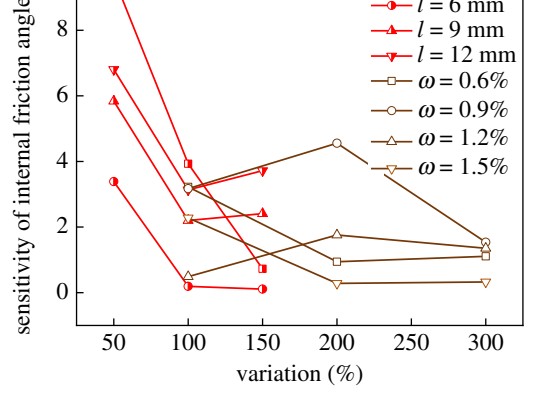

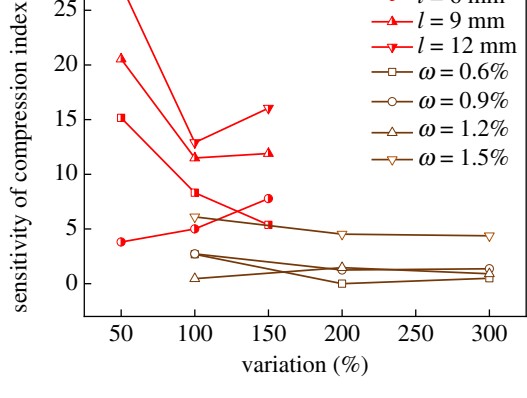

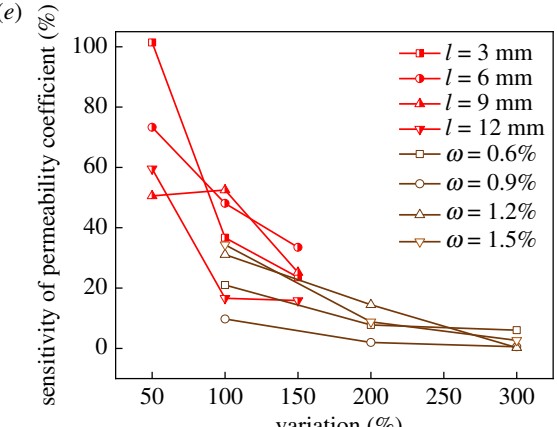

**Figure 9.** Sensitivity analysis: (*a*) shear strength; (*b*) cohesion; (*c*) internal friction angle; (*d*) compression index; and (*e*) permeability coefficient.

# 4. Micro imaging explication of BFRT

The shear strength of tailings dam plays a key role in resisting failure, and the mechanical properties of tailings have great influence on tailings dam shear strength. In order to investigate the intrinsic mechanism of BF improving tailings strength, the SEM experiment was carried out [39]. The previous experiment results indicate that shear strength and cohesion of BFRT are positively correlated with fibre content under the same length, the reason being that when there is no BF in tailings, reliance is mainly on the friction between tailings particles and matrix suction to resist shear failure. As shown in figure 10*a*, when $\omega$ is small, fibres are scattered in tailings, and the specimen would be damaged only if it overcomes the friction between tailings particles and the friction between fibre

(a)

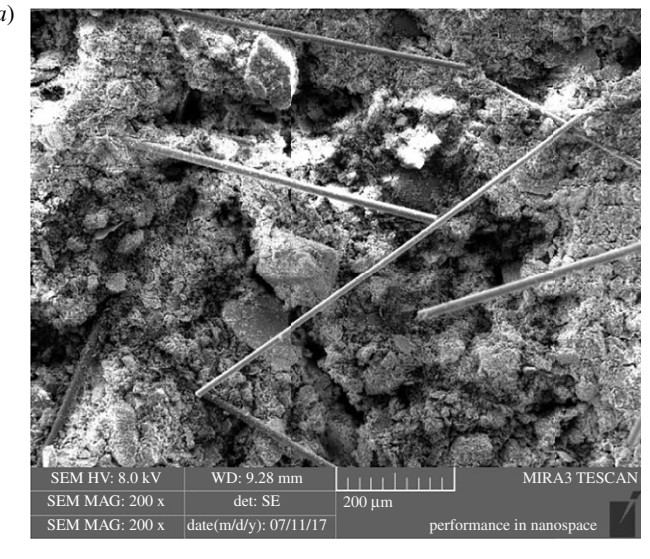

(b)

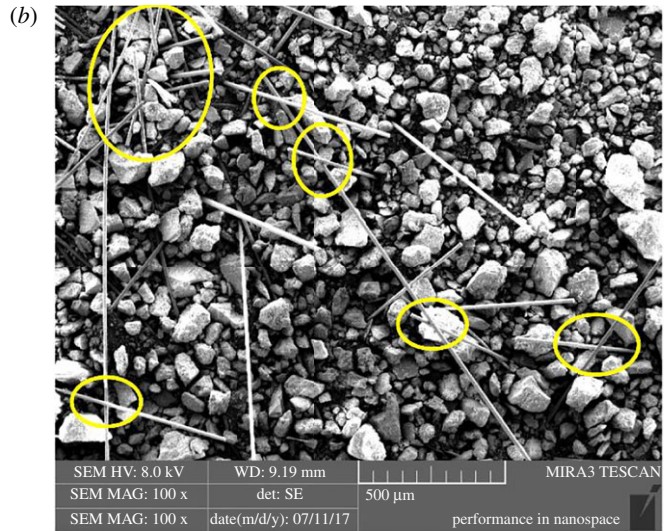

(c)

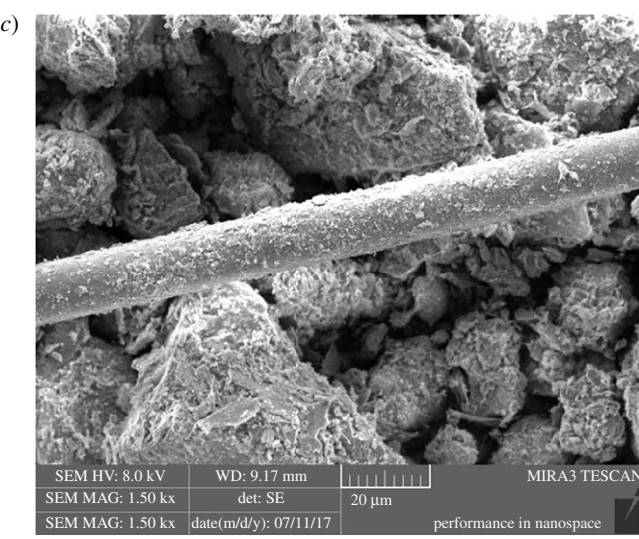

**Figure 10.** SEM images of BFRT: (a) scattered basalt fibres; (b) network structure; and (c) basalt fibre monofilament.

monofilaments and tailings particles, namely BF enhances the interaction between tailings particles. On the other hand, the tensile strength and fracture strength of BF monofilaments are much greater than those of tailings particles, which contributes to boost the shear strength. Figure 10b demonstrates that

the amount of BF per unit volume increases with increasing $\omega$; the fibre monofilaments connect with each other to form a network structure, the network wraps tailings particles up to form a whole body, which strengthens the interfacial force between fibre monofilaments and tailings particles [40]. Thus the cohesion and shear strength are improved, which is why shear strength and cohesion of BFRT are positively correlated with fibre content under the same length to some extent. When fibre length is too long and exceeds a certain value, while the sample size is fixed ($\varphi = 61.8$ mm, $H = 20$ mm), the longer fibre has greater probability of non-uniform distribution in BFRT under the same dosage, resulting in fibre local concentration and fibre clusters [20], even forming weak structural planes between tailings particles and weakening the matrix suction and friction, thus the integrity of the specimen is destroyed, which leads to shear strength and cohesion decrease. That is why it is not that the longer the fibre is, the greater the shear strength and cohesion are, but there is an optimal length of BFRT. BF monofilament surface is smoother compared with the roughness of tailings particles (figure 10$c$) and the shape of tailings particles varies dramatically, so BF is not able to significantly alter tailings particle roughness, thereby it has little effect on internal friction angle. Consequently, the intrinsic reason for shear strength increase of BFRT is the cohesion elevation caused by BF.

# 5. Conclusion

Using BF for tailings reinforcement has hitherto been rarely studied. In order to study its effect on the strength, consolidation and permeation characteristics of tailings, laboratory experiments were conducted in this investigation to provide necessary guidance for the application of BF reinforcement tailings into engineering practice. According to the experimental research work in this paper, the conclusions are as follows:

(1) Adding BF into tailings can enhance the shear strength and cohesion, but has little effect on the internal friction angle. When $l$ is constant, $\tau$ and $c$ are positively correlated with $\omega$; while $\omega$ is constant, $\tau$ and $c$ do not constantly increase with the increase of $l$ but ascend to the peak firstly and then descend as $l$ increases, and $\tau$ obtains the maximum under $l = 6$ mm, while $c$ acquires the maximum under $l = 6$–$9$ mm.

(2) The addition of BF can enhance tailings compression index. $l$ variation has no significant effect on $C_c$ under the same dosage, but it is positively correlated with $\omega$ under the same length; 0.6% is the critical content that leads to BFRT permeability coefficient variation, and $k$ is less than that of original tailings when $\omega = 0.6\%$, while it is invariably greater when $\omega > 0.6\%$.

(3) The sensitivity of each index to BF content is greater than that of length, and the sensitivity is ranked as follows: $c > k > \tau > C_c > \varphi$.

The experiment results promote the understanding of BFRT mechanical properties. They suggest that it is technically feasible to improve tailings mechanical properties by BF reinforcement, which provides another potential way for improving the safety of tailings dam and other artificial soil slopes or soil structures.

Data accessibility. The data are provided in the electronic supplementary material [41].
Authors' contributions. Conceptualization, H.Y. and D.Z.; funding acquisition, J.L. and D.Z.; methodology, H.Y. and B.Z.; supervision, J.L., Y.W. and D.Z.; validation, J.L., D.Z., W.X. and Y.Y.; writing—original draft, H.Y.; writing—review and editing, J.L., D.Z., W.X., C.Y. and Y.Y. All authors have read and approved the final manuscript.
Competing interests. We declare we have no competing interests.
Funding. This work was supported by the Scientific Research Foundation of State Key Laboratory of Coal Mine Disaster Dynamics and Control (grant no. 2011DA105287-zd201804), the National Natural Science Foundation of China (grant no. 51874053) and the Fundamental Research Funds for the Central Universities (project no. 2020CDCGJ041).
Acknowledgements. We thank Dr Xiaolei Wang for his comments on the revision of the manuscript.

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
