## [Peer Review File · Royal Society Open Science]

Review History

RSOS-210669.R0 (Original submission)

Review form: Reviewer 1

Is the manuscript scientifically sound in its present form?

Yes

Are the interpretations and conclusions justified by the results?

Yes

Is the language acceptable?

Yes

Do you have any ethical concerns with this paper?

No

Have you any concerns about statistical analyses in this paper?

No

Recommendation?

Accept with minor revision (please list in comments)

Comments to the Author(s)

The authors conducted laboratory tests of adding Basalt Fiber (BF) into tailings for reinforcement to study the strength, consolidation and permeation characteristics of Basalt Fiber Reinforced Tailings (BFRT). In addition, the sensitivity of different fiber lengths and contents on shear strength, cohesion, internal friction angle, compression index and permeability coefficient of BFRT were analyzed. Finally, the SEM was used to investigate the intrinsic mechanism of basalt fiber reinforcement to improve tailings strength. The work is interesting, but some minor revisions are needed before publication:

1. Eliminating the redundant citations and only keep the necessary ones, such as citations from 34 to 37.
2. In figures 3 and 4a, some characters are too small to distinguish
3. In lines 168 and 169, it would be better to define 'optimal length' and 'optimal content' at first.
4. How to obtain the shear strength, cohesion, internal friction angle in section 3.1.1?
5. In section 4, the author illustrated the mechanism of BF to improve the strength of tailings. However, the corresponding citations are missing.
6. In keywords, 'tit' would be more appropriate to modify 'shear strength' to 'strength'.

Review form: Reviewer 2

Is the manuscript scientifically sound in its present form?

Yes

Are the interpretations and conclusions justified by the results?

Yes

Is the language acceptable?

No

Do you have any ethical concerns with this paper?

No

Have you any concerns about statistical analyses in this paper?

Yes

Recommendation?

Accept with minor revision (please list in comments)

Comments to the Author(s)

Analysis and conclusions drawn are well thought out and complete. Just need to clean up some of the language as there are a lot of run on sentence and poor sentence structure within the paper that needs to be addressed.

Decision letter (RSOS-210669.R0)

Dear Dr Wang

On behalf of the Editors, we are pleased to inform you that your Manuscript RSOS-210669 "Mechanical and permeation response characteristics of basalt fiber reinforced tailings to different reinforcement technologies: an experimental study" has been accepted for publication in Royal Society Open Science subject to minor revision in accordance with the referees' reports. Please find the referees' comments along with any feedback from the Editors below my signature.

Please submit your revised manuscript and required files (see below) no later than 7 days from today's (ie 09-Aug-2021) date. Note: the ScholarOne system will 'lock' if submission of the revision is attempted 7 or more days after the deadline. If you do not think you will be able to meet this deadline please contact the editorial office immediately.

on behalf of Professor Zach Agioutantis (Associate Editor) and R. Kerry Rowe (Subject Editor)
openscience@royalsociety.org

Reviewer comments to Author:
Reviewer: 1

Comments to the Author(s)

The authors conducted laboratory tests of adding Basalt Fiber (BF) into tailings for reinforcement to study the strength, consolidation and permeation characteristics of Basalt Fiber Reinforced Tailings (BFRT). In addition, the sensitivity of different fiber lengths and contents on shear strength, cohesion, internal friction angle, compression index and permeability coefficient of BFRT were analyzed. Finally, the SEM was used to investigate the intrinsic mechanism of basalt

fiber reinforcement to improve tailings strength. The work is interesting, but some minor revisions are needed before publication:

1. Eliminating the redundant citations and only keep the necessary ones, such as citations from 34 to 37.
2. In figures 3 and 4a, some characters are too small to distinguish
3. In lines 168 and 169, it would be better to define 'optimal length' and 'optimal content' at first.
4. How to obtain the shear strength, cohesion, internal friction angle in section 3.1.1?
5. In section 4, the author illustrated the mechanism of BF to improve the strength of tailings. However, the corresponding citations are missing.
6. In keywords, it would be more appropriate to modify 'shear strength' to 'strength'.

Reviewer: 2

Comments to the Author(s)

Analysis and conclusions drawn are well thought out and complete. Just need to clean up some of the language as there are a lot of run on sentence and poor sentence structure within the paper that needs to be addressed. (Please see attached RSOS-210669_Proof_hi.pdf)

===PREPARING YOUR MANUSCRIPT===

===PREPARING YOUR REVISION IN SCHOLARONE===

To revise your manuscript, log into <https://mc.manuscriptcentral.com/rsos> and enter your Author Centre - this may be accessed by clicking on "Author" in the dark toolbar at the top of the

page (just below the journal name). You will find your manuscript listed under "Manuscripts with Decisions". Under "Actions", click on "Create a Revision".

<https://royalsociety.org/journals/authors/author-guidelines/#supplementary-material> to include a suitable title and informative caption. An example of appropriate titling and captioning may be found at https://figshare.com/articles/Table_S2_from_Is_there_a_trade-off_between_peak_performance_and_performance_breadth_across_temperatures_for_aerobic_sc_ope_in_teleost_fishes_/3843624.

Author's Response to Decision Letter for (RSOS-210669.R0)

See Appendix A.

Decision letter (RSOS-210669.R1)

Dear Dr Wang,

I am pleased to inform you that your manuscript entitled "Mechanical and permeation response characteristics of basalt fiber reinforced tailings to different reinforcement technologies: an experimental study" is now accepted for publication in Royal Society Open Science.

on behalf of Professor Zach Agioutantis (Associate Editor) and R. Kerry Rowe (Subject Editor)
openscience@royalsociety.org

Appendix A

Change of authorship

Dear editor,

Thank you very much for processing our manuscript, we deeply appreciate your hard work and there's no way we can ever thank you enough. We feel sorry to bother you, we write this letter to apply for the change of authorship and we hope it won't cause you much inconvenience.

During the manuscript revision process, we received a lot of constructive comments and advices from Prof. Jianzhong Liu, and Prof. Liu participated in the writing and editing of the revised manuscript in the whole process, Prof. Liu has made a lot of contributions to the revision of this paper. Simultaneously, Prof. Liu also provided parts of the financial supports for our research, hence he has qualified as an author of the paper per the guidelines of *Royal Society Open Science* at <https://royalsociety.org/journals/ethics-policies/openness/>. Therefore, after careful consideration by all authors, we all consent to add Prof. Jianzhong Liu as one of the author of this paper and reorder the existing authors.

In addition, Prof. Dongming Zhang participated in the conceptualization, writing, editing, revision, supervision and validation of this paper, and he undertook most of the financial supports of this research, he has made a lot of contributions to this paper. Hence all authors agree to change Dr. Yun Wang to Prof. Dongming Zhang as the corresponding author. Henceforth, Prof. Dongming Zhang takes in charge of contacting with the editorial office and other subsequent affairs. The email of Dongming Zhang is: zhangdm@cqu.edu.cn. The details are listed below:

Manuscript number: RSOS-210669.

Manuscript title: Mechanical and permeation response characteristics of basalt fiber reinforced tailings to different reinforcement technologies: an experimental study.

Indicate the specific change:

Adding a new author

Name	Jianzhong Liu
Email address	liujianzhong@ccteg.cn
Institution	State Key Laboratory of Coal Mine Disaster Dynamics and Control, Chongqing University; School of Resources and Safety Engineering, Chongqing University; China Coal Technology & Engineering Group
Specific contribution	Revision, Writing, Editing, Validation, Supervision, Funding

Change of corresponding author

Name	Dongming Zhang
Email address	zhangdm@cqu.edu.cn
Institution	State Key Laboratory of Coal Mine Disaster Dynamics and Control, Chongqing University; School of Resources and Safety Engineering, Chongqing University
Specific contribution	Conceptualization, Writing, Editing, Revision, Supervision, Validation, Funding

Complete author order AFTER change

Order	Author name BEFORE change	Author name AFTER change	Signature
1	Han Yang	Jianzhong Liu	Jianzhong Liu
2	Yun Wang	Han Yang	Han Yang
3	Dongming Zhang	Dongming Zhang*	Dongming Zhang
4	Weijing Xiao	Yun Wang	Yun Wang
5	Chen Ye	Weijing Xiao	Weijing Xiao
6	Binbin Zheng	Chen Ye	Chen Ye
7	Yushun Yang	Binbin Zheng	Binbin Zheng
8		Yushun Yang	Yushun Yang

Authorship statement

All authors declare that:

1) The manuscript is not currently under consideration, in press, or published elsewhere, and the research reported will not be submitted for publication elsewhere until a final decision has been made as to its acceptability by the journal.

2) The manuscript is truthful original work without fabrication, fraud, or plagiarism.

3) All authors have made important scientific contributions to the study and thoroughly familiar with the primary data. And all authors have read the complete manuscript and take responsibility for the content and completeness of the manuscript.

Competing interests:

The authors assure that they have no competing interests. All authors have read and approved the final authorship and manuscript.

Appendix B

Response to editor and reviewers' comments

Dear Editor,

Thank you very much for your help in processing the review of our manuscript (**Manuscript ID: RSOS-210669**). We deeply appreciate your hard work and there's no way we can ever thank you enough. We have carefully read the thoughtful comments from you and reviewers and found that these suggestions are helpful for us to improve our manuscript. On the basis of the enlightening questions and helpful advices, we have now completed the revision of our manuscript, and all revisions have been highlighted in the attached file named “**Revised manuscript—Highlight version**”. The contents in pink in the highlight version manuscript are what we added, the contents in turquoise with strikethrough are what we deleted. We hope that all these corrections and revisions would be satisfactory. If you have any questions at all, please do not hesitate to contact the corresponding author Dongming Zhang whose E-mail is zhangdm@cqu.edu.cn. Thanks a lot, again! The itemized responses to the reviewers' comments are listed below:

Reviewer 1 comments:

The authors conducted laboratory tests of adding Basalt Fiber (BF) into tailings for reinforcement to study the strength, consolidation and permeation characteristics of Basalt Fiber Reinforced Tailings (BFRT). In addition, the sensitivity of different fiber lengths and contents on shear strength, cohesion, internal friction angle, compression index and permeability coefficient of BFRT were analyzed. Finally, the SEM was used to investigate the intrinsic mechanism of basalt fiber reinforcement to improve tailings strength. The work is interesting, but some minor revisions are needed before publication:

1. Eliminating the redundant citations and only keep the necessary ones, such as citations from 34 to 37.

Response: We have deleted redundant references and only kept the necessary ones as suggested by reviewer, please see line 83.

2. In figures 3 and 4a, some characters are too small to distinguish.

Response: Figure 3 and figure 4 have been edited carefully and the characters have been enlarged for easier and clearer identification. The modifications are shown in figure 3 (line

153) and figure 4 (line 165) respectively.

3. In lines 168 and 169, it would be better to define 'optimal length' and 'optimal content' at first.

Response: We have rearranged the order of the paragraphs, we deleted parts of contents in line 205 ~208, and defined 'optimal length' and 'optimal content' in the first paragraph below figure 4 in line 168~173. In addition, some necessary substances have been added and the sentences was reorganized, the modifications have been highlighted in the revised manuscript in section 3.1.1.

4. How to obtain the shear strength, cohesion, internal friction angle in section 3.1.1?

Response: The max shear stress of τ - Δl curve is taken as the shear strength, and when there is no maximum, the corresponding shear stress when shear displacement equals 4 mm is taken as the shear strength. According to Coulomb's law, the shear strength curve τ - σ can be plotted with τ as the ordinate and the vertical stress σ as the abscissa, the cohesion and internal friction angle of BFRT can be calculated accordingly. We have made detailed explanations and added those contents to the revised manuscript in first paragraph in section 3.1.1, please see line 147~151.

5. In section 4, the author illustrated the mechanism of BF to improve the strength of tailings. However, the corresponding citations are missing.

Response: We've checked this section and add necessary references in line 425 and line 430 as suggested by reviewer.

6. In keywords, it would be more appropriate to modify 'shear strength' to 'strength'.

Response: We have revised 'shear strength' to 'strength' in keywords as suggested by reviewer, please see line 28.

Reviewer 2 comments:

Analysis and conclusions drawn are well thought out and complete. Just need to clean up some of the language as there are a lot of run on sentence and poor sentence structure within the paper that needs to be addressed. (Please see attached RSOS-210669_Proof_hi.pdf).

Response: Since reviewer 2 directly annotated the comments in the manuscript, hence we followed the reviewer's comments and made revisions point-by-point according to the comments directly, the corresponding modifications have been highlighted in the revised

manuscript. Specific revisions are listed below:

1. In line 37, “it’s” has been changed to “It’s” as suggested by reviewer.
2. In line 38, 18000 and 7800 has been changed to 18,000 and 7,800 respectively as suggested by reviewer.
3. We’ve deleted the contents in line 41~42, and explained the term “high potential energy” in detail, corresponding explanations have been added in line 42~44.
4. Case studies have been added as suggested by reviewer, please see line 44~50.
5. We’ve made detailed instructions on “reinforced earth theory”, corresponding explanations have been added in line 67~71.
6. “figure” has been changed to “Figure” in line 160, “chapters” has been changed to “sections” in line 163.
7. Sentences have been re-edited in line 174~192, and the semicolon has been deleted, the modifications were shown in line 193~204.
8. “the” has been revised to “The” in line 179, “but” has been deleted in line 222, and the sentences have been re-edited in line 217~228.
9. The sentences in line 229~233 have been deleted, and the re-edited sentences were added in line 234~235 as suggested by reviewer.
10. Sentences have been re-edited in line 239~261.
11. Corresponding explanations about equation (3.1) have been made, and the sentences have been re-edited, please see line 263~278. We have also modified the format of other formulas, please see line 334, line 337 and line 381.
12. We’ve re-write the paragraphs below figure 7, and the conclusions have been re-written too, please see line 279~326.
13. Sentences have been re-edited and corresponding instructions about seepage channels have been made, please see line 358~377.
14. Function (3.4) has been re-edited, and corresponding conclusions have been re-written, please see line 379~409.
15. The word “indoor” has been changed to “laboratory”, please see line 445.

16. We've re-written the last paragraph and made corresponding supplementary instructions in section 5, please see line 461~467.

Additional modifications:

In addition to the modifications made according to the comments of editors and reviewers, we carefully checked the whole manuscript and made some additional modifications. The specific modifications are listed below:

1. The authorship has been changed, Jianzhong Liu and corresponding address information has been added, the corresponding author has been changed and the E-mail information has been modified, please see line 4~15. (Note: the *change of authorship* statement has been uploaded as an attachment).
2. "thus" has been revised to "Thus", please see line 33.
3. Some sentences have been re-edited for better expression, please see line 52~54, line 80~91, line 105~109, line 114~119, line 124~138, line 159~163, line 205~209, line 346~357 and line 421~426.
4. Some words have been amended for more appropriate expression, please see line 99~102, line 159~163, line 329~333, line 341, line 415, line 419, line 431~437 and line 444.
5. The "Authors' contributions" section has been re-edited, please see line 472~474.
6. The *Scientific Research Foundation of State Key Laboratory of Coal Mine Disaster Dynamics and Control (Grant Number: 2011DA105287-zd201804)* has been added in the "Funding" section, because it provided financial support for our research, and the recipient is the new corresponding author Dongming Zhang, please see line 478~479.
7. The "Acknowledgements" section has been added before references, please see line 482~483
8. The "References" section has been re-edited, redundant citations have been deleted and the necessary ones have been added. Also, the orders of those references have been modified to the correct order. please see line 495~601.